# Limus Devices for the Treatment of SFA: Latest Outcomes and Future Perspectives

**DOI:** 10.3390/jcm14103594

**Published:** 2025-05-21

**Authors:** Genti Xhepa, Agostino Inzerillo, Ilinca Constantinescu, Pierre Faerber, Adrien Gleyzolle, Pierpaolo Biondetti, Filippo Del Grande, Edon Xhepa, Simone Mortellaro, Gianpaolo Carrafiello, Giuseppe Pellegrino, Alexis Ricoeur

**Affiliations:** 1Istituto di Imaging della Svizzera Italiana (IIMSI), Ente Ospedaliero Cantonale (EOC), 6900 Lugano, Switzerland; genti.xhepa@eoc.ch (G.X.); filippo.delgrande@eoc.ch (F.D.G.); 2Interventional Radiology Unit, University Hospital of Geneva (HUG), 1205 Geneva, Switzerland; ilinca.constantinescu@hug.ch (I.C.); pierre.faerber@hug.ch (P.F.); adrien.gleyzolle@hug.ch (A.G.); alexis.ricoeur@hug.ch (A.R.); 3AOUP Paolo Giaccone, Biomedicine, Neuroscience and Advanced Diagnostic Department (BiND), University of Palermo, 90127 Palermo, Italy; inzerilloagostino@gmail.com; 4Department of Radiology, Foundation IRCCS Ca’ Granda-Ospedale Maggiore Policlinico, University of Milan, 20122 Milan, Italy; pierpaolo.biondetti@policlinico.mi.it (P.B.); gianpaolo.carrafiello@unimi.it (G.C.); 5Facoltà di Scienze Biomediche, Campus Est, Università Della Svizzera Italiana (USI), 6900 Lugano, Switzerland; 6Postgraduate School in Radiodiagnostics, University of Milan, 20122 Milan, Italy; simone.mortellaro@unimi.it (S.M.); giuseppe.pellegrino@unimi.it (G.P.); 7Department of Oncology and Hemato-Oncology, University of Milan, 20122 Milan, Italy

**Keywords:** sirolimus, paclitaxel, drug-eluting stent, drug-eluting balloon, superficial femoral artery, popliteal artery, bioresorbable scaffolds, endovascular treatment

## Abstract

Globally, cardiovascular disease is a leading cause of disability and early death, affecting 422.7 million people and causing 17.9 million deaths (31% of global deaths) in 2015. Peripheral arterial disease, previously overlooked compared to coronary artery disease, is now recognised as a major contributor to cardiovascular morbidity and mortality, with distinct characteristics. After noninvasive methods, the femoropopliteal segment is frequently treated with revascularisation, which is recommended for claudication and chronic limb-threatening ischemia (CLTI). Challenges such as mechanical stresses, chronic occlusions, extensive plaque, and calcification affect procedural success and vessel patency. Innovations were needed to address these issues, and vascular drug delivery devices have become integral to endovascular treatment. We review the current literature concerning a diverse range of these devices in clinical use and their role in managing symptomatic patients.

## 1. Introduction

Globally, cardiovascular disease is a major cause of disability and premature death. In 2015, it affected about 422.7 million people and caused 17.9 million deaths, accounting for 31% of global deaths. Notably, cardiovascular diseases have significantly impacted low- and middle-income countries, where over 80% of related deaths occur [1,2,3,4]. Projections indicate that by 2030, approximately 23.6 million people may die annually from cardiovascular diseases [5]. Additionally, in the United States, the prevalence of peripheral artery disease (PAD) varies among different demographic groups, leading to disparities in disease burden [4] A study analysing CDC WONDER data from 2000 to 2019 found that, while age-adjusted mortality rates for peripheral artery disease initially declined, this trend plateaued after 2016, with over 1.9 million PAD-related deaths recorded. Marked disparities persisted, disproportionately affecting men, non-Hispanic Black individuals, adults aged 85 and older, and rural populations. Additionally, a concerning upturn in crude mortality emerged among individuals aged 25–39 in the last decade, underscoring ongoing inequities and the necessity for targeted public health strategies (Issa et al. 2023) [6].

While PAD has historically received less attention than coronary artery disease (CAD) and cerebrovascular disease, it is now acknowledged as a significant contributor to cardiovascular morbidity and mortality. Lower extremity PAD involves narrowing or blockage in the arteries from the aortoiliac segment to the pedal arteries. Although there are shared risk factors with other forms of atherosclerosis, recent epidemiological data emphasise the distinct nature of PAD [7]. The femoropopliteal segment is the most frequently treated region for symptomatic PAD patients. Revascularisation is recommended for claudication that limits daily activities and chronic limb-threatening ischemia (CLTI) after noninvasive approaches have been exhausted [8,9].

Endovascular intervention in the femoropopliteal segment was pioneered by Charles Dotter in 1964 [10], initially utilising coated dilators for angioplasty. Since then, there has been a significant advancement in the variety and sophistication of available devices. Despite these developments, practical challenges—such as the femoropopliteal artery’s susceptibility to mechanical stresses (including flexion, extension, compression, elongation, and torsion), the high incidence of chronic total occlusions, extensive atherosclerotic plaque, and severe vascular calcification—continue to compromise procedural success and long-term vessel patency. The field has seen rapid innovation in endovascular technologies to address these limitations. Drug-eluting devices have become especially prominent, offering targeted delivery of antiproliferative agents to minimise neointimal hyperplasia and restenosis. Drug-eluting stents (DES) and drug-eluting balloons (DEB), primarily utilising paclitaxel due to its lipophilic properties and sustained tissue retention, have demonstrated clinical efficacy in reducing restenosis rates compared to plain balloon angioplasty or bare-metal stents. More recently, bioresorbable drug-eluting scaffolds represent a promising technology designed to provide temporary luminal support and drug delivery, followed by gradual resorption to restore natural vessel function [10], using coated dilators for angioplasty. Over time, the array of available devices has expanded. However, practical challenges such as mechanical stresses (flexion/extension, compression/elongation, and torsion), high rates of chronic total occlusions, extensive plaque, and significant calcification continue to affect procedural success and long-term vessel patency. Innovation in this area is in high demand, with several purpose-built devices addressing these issues now available or nearing launch [8,9,11,12].

### 1.1. TAXUS Versus LIMUS, Biochemical Consideration

The two primary pharmacologic classes used are taxanes and limus drugs [13]. Taxanes, such as paclitaxel, being commonly used with litaxel, exert their effects by binding to and disrupting microtubules, essential structures during cell mitosis, which leads to cell death and inhibition of proliferation—an effective antitumor mechanism. Paclitaxel is highly lipophilic, allowing it to persist in tissue for extended periods, although it can become cytotoxic and apoptotic at high doses, presenting its main limitations. Sirolimus forms an immunosuppressive complex with FKBP12, inhibiting mTOR and causing cell-cycle arrest at G1 and S phases. Although less lipophilic than paclitaxel, sirolimus has a broader therapeutic range, making it advantageous [14,15,16]. Sirolimus effectively suppresses cell proliferation in animal studies, while paclitaxel tends to have the opposite effect [17]. In normoxic conditions, both drugs show similar inhibitory effects on cell proliferation. Still, under hypoxia, sirolimus maintains its anti-proliferative effect. At the same time, paclitaxel’s effectiveness decreases, especially at high doses, which carry a higher risk of systemic toxicity due to paclitaxel’s narrow therapeutic range [18]. Sirolimus is evenly distributed across all layers of the vessel wall. In contrast, paclitaxel predominantly accumulates in the adventitia with a lower transmural gradient [13].

### 1.2. Review Method

A structured and transparent literature review was conducted to identify studies involving percutaneous treatment of the superficial femoral artery (SFA) and/or popliteal artery, with a focus on clinical outcomes, follow-up data, and complications, including only full-text publications in English from January 2000 to September 2023. Based on these predetermined inclusion criteria, a systematic search of PubMed (MEDLINE) and Google Scholar was performed in October 2023 using both controlled vocabulary and free-text terms, specifically the keywords “Sirolimus” OR “limus” AND (“drug eluting stent” OR “drug eluting balloon” OR “resorbable scaffolds” OR “drug eluting device”), tailored to the main thematic areas of this review. Reference lists of eligible articles were hand-searched to identify additional studies not captured in the initial search. All retrieved records were imported into citation management software, and duplicate articles were removed before screening. Two reviewers independently screened the titles and abstracts of all studies according to the predefined criteria, with potentially relevant studies advancing to full-text review. Any disagreements regarding inclusion were resolved by consensus, and where necessary, a third senior reviewer was consulted to provide further guidance. Data extraction was performed independently by both reviewers using a standardised data collection form to ensure accuracy and consistency. Given the considerable heterogeneity in study designs, outcome measures, case report predominance, and inconsistent follow-up intervals, meta-analysis was not performed and a narrative synthesis approach was adopted. Ultimately, a total of 15 studies fulfilled all eligibility criteria and were included in this review. To enhance transparency and reproducibility, a PRISMA flowchart depicting the study selection process is now included. The findings and methodology of this review are informed by previously published studies detailing epidemiological context and disparities in peripheral artery disease [1,2,3,4], as well as original research on endovascular and percutaneous interventions for the femoropopliteal segment utilising a range of devices and techniques (Figure 1) [19,20,21,22].

### 1.3. Sirolimus Devices

#### 1.3.1. STENT—Historical Data

The SIROCC O trial (“Sirolimus-Coated Cordis S.M.A.R.T. Nitinol Self-Expandable Stent for the Treatment of Obstructive Superficial Femoral Artery Disease”) evaluated sirolimus-eluting S.M.A.R.T. stents for treating superficial femoral artery (SFA) disease in a randomised, double-blind study at eight centres in Europe and Australia. Patients received either a sirolimus-eluting or bare stent, with the former containing 90 µg/cm^2^ sirolimus within a 5 µm thick copolymer matrix. Both groups showed comparable results in peak systolic velocity ratios, Rutherford classification, ankle brachial index (ABI) improvement, and in-stent restenosis at 24 months post-implantation [23].

The STRIDES trial assessed the safety and performance of an everolimus-eluting self-expanding stent system in a prospective, nonrandomised, single-arm study at 11 European centres from May 2007 to January 2008. The stent, called “Dynalink-E,” combined the Dynalink nitinol self-expanding stent, everolimus, and an EVAL copolymer, with a total drug load of 225 µg/cm^2^ stent surface area. It had a prolonged drug release profile, with about 80% released slowly over 90 days, unlike coronary stents designed for 30-day release [24].

Notably, the STRIDES trial experienced a high number of restenotic events between months 7 and 12. The 94% primary patency rate at 6 months decreased to 68% at 1 year. A similar trend was observed in the SIROCCO trial, where the 5% restenosis rate at 6 months increased to 19% at 18 months, comparable to the rate in bare nitinol controls. This suggests that none of these drug-eluting stents had a long enough drug release profile to address ongoing interactions between the stent and SFA [23,24].

#### 1.3.2. NiTiDES Stent

The NiTiDES stent, created by CID S.p.a. (a member of the Alvimedica Group), is a CE-certified, polymer-free, self-expanding, drug-eluting stent made of nitinol alloy. It contains the amphilimus formulation (sirolimus plus fatty acid) to enhance drug bioavailability, with the drug located in grooves (reservoirs) on the stent’s outer surface, using Abluminal Reservoir Technology. This design ensures drug elution only toward the vessel wall, promoting rapid re-endothelialisation of the stent. To enhance hemocompatibility, biocompatibility, and thromboresistance, the entire structure, including the reservoirs, is uniformly coated with an ultra-thin layer of pure carbon known as i-Carbofilm (Bio Inducer Surface) [25].

The first-in-human prospective, multicentre, single-arm study (ILLUMINA = Innovative siroLimus self Expanding drUg-eluting Stent for the treatMent of perIpheral Disease: Evaluation of Safety aNd efficAcy) involved 100 patients with de novo or restenotic native lesions to evaluate the effectiveness and safety of the NiTiDES stent. At 12 and 24 months, the primary patency rates were 89.3% and 83.4%, respectively. Freedom from clinically driven target lesion revascularisation (CD-TLR) remained at 93.1% throughout 24 months. By the 24-month time point, 86.9% of patients reported an improvement in Rutherford clinical category from baseline. Functional benefits persisted, with significant enhancements in Walking Impairment Questionnaire scores (WIQ) and ABI measurements compared to baseline [26].

### 1.4. Bioresorbable Vascular Scaffold

A bioresorbable vascular scaffold (BVS) provides temporary radial support during acute procedures and healing phases, addressing issues such as flow-limiting dissection and vascular recoil. Unlike metal stents, BVS dissolves naturally, avoiding long-term problems such as vessel wall irritation, fractures, and imaging artefacts (e.g., in computed tomography angiography and magnetic resonance angiography) [26,27,28,29].

Two primary types of bioresorbable stent materials are currently used: magnesium-based stents, which gradually degrade in the vascular wall, releasing non-toxic magnesium and inorganic salts until complete dissolution, and polymer poly-L-lactic acid (PLLA)-based stents which degrade through hydrolysis, converting into lactic acid, further breaking down into carbon dioxide and water. Recent experience with bioresorbable stents in peripheral artery disease (PAD) has been gained through initial clinical trials and ongoing studies, raising questions about their readiness for large randomised controlled trials (RCTS) in PAD [30,31].

The ESPRIT BVS system (Abbott Vascular, Santa Clara, California) consists of a poly-L-lactide polymer backbone coated with a drug-eluting matrix containing 100 µg of everolimus/cm^2^ of scaffold. It features a design with expandable serpentine rings connected by links, with platinum markers at each end for fluoroscopic visibility. The ESPRIT BVS exhibits crush recovery properties, enabling it to withstand mechanical forces in the SFA [32].

The ESPRIT I clinical study was a prospective, single-arm, open-label, multicentre trial conducted at seven clinical sites with 35 enrolled subjects. Results at 1 and 2 years showed low restenosis rates of 12.1% and 16.1%, respectively, along with low rates of TLR at 8.8% and 11.8%. Improvement in the Rutherford–Becker clinical category and walking distance was sustained up to 24 months of follow-up. Most lesions were in the superficial femoral artery (SFA) at 88.6%, followed by the external iliac artery (EIA) at 11.4%. The lesion length was 35.7 ± 16.0 mm, with no severe calcifications reported by investigators [32].

### 1.5. Drug-Eluting Ballon

#### Magic Touch

The MagicTouch PTA is a sirolimus-coated balloon (SCB) that utilises proprietary nanolute technology to enhance the effectiveness of sirolimus. Sirolimus is transformed into submicron-sized particles and enclosed within phospholipid-drug nanocarriers, improving its adhesion to the balloon surface at a dose of 1.27 μg/mm^2^. When the MagicTouch PTA is inflated at the treatment site, the highly biocompatible submicron sirolimus carrier is transferred to the vessel wall through coefficient diffusion. Because of its small size, sirolimus can easily penetrate the artery’s adventitial layers and, as the body’s pH changes, the submicron carrier mimics body lipids and releases sirolimus, effectively preventing restenosis in target arteries [33].

The XTOSI FIH pilot study is a single-centre trial assessing the MagicTouch PTA SCB. It includes 50 patients, with 20 receiving SCB treatment for SFA disease. The primary endpoint, defined by duplex ultrasound with a peak systolic velocity ratio (PSVR) of ≤2.4, was measured to assess primary patency at 6 months. The vast majority (90%) of patients had CLTI. The treated SFA lesions had an average length of 277 mm, with 45% being chronic total occlusions. Primary patency rates for femoropopliteal vessels at 6, 12, and 24 months were 88%, 79%, and 53%, respectively. For below-the-knee (BTK) vessels, primary patency rates at these time points were 74%, 59%, and 50%. At 24 months, the study reported 89% freedom from TLR, 76% amputation-free survival, 18% all-cause mortality, 92% limb salvage success, and 100% wound healing in survivors with intact limbs [33,34].

In the next few years, we will have the results of some interesting studies which will shed light on the performance of this device.

The Sirpad trial is a Phase III randomised controlled trial initiated by investigators. It aims to determine if sirolimus-coated balloon angioplasty is at least as effective and potentially better than uncoated balloon angioplasty in adults with infra-inguinal peripheral arterial disease requiring endovascular angioplasty. The primary efficacy outcome includes major amputation of the target limb and revascularisation of the target lesion within one year. The primary safety outcome is all-cause mortality. This study will provide valuable information about sirolimus-coated balloon catheters’ effectiveness and safety in a diverse patient population (targeting 1200 patients). Due to safety concerns with paclitaxel-coated devices, mortality data will be collected for up to 5 years after randomisation [35,36].

Similarly, the FUTURE SFA (Randomized Controlled Trial of First SirolimUs CoaTed Balloon VersUs StandaRd Balloon Angioplasty in the TrEatment of Superficial Femoral Artery and Popliteal Artery Disease) trial is a double-blind, multicentre, randomised controlled trial comparing sirolimus drug-coated balloons to standard angioplasty for treating superficial and popliteal arterial disease. It will include 153 patients with PAD (Rutherford class 3 to 6), with most having CLTI, which is a common indication in Asia. Patients will be randomly assigned in a 2:1 ratio to receive either the Magic Touch PTA or standard balloon angioplasty. The primary outcome is six-month primary patency, defined as a duplex PSVR of 2.4 or less (without the need for target lesion revascularisation) [37].

Assessment of Sirolimus- vs. paCLitaxEl-coated balloon angioPlasty In atherosclerotic femoropopliteal lesiOnS (ASCLEPIOS Study) is a prospective, randomised, controlled, single-centre, noninferiority study comparing MagicTouch versus Ranger tm. It focuses on the primary outcomes of procedural success and primary vessel patency at the index procedure. Secondary outcomes include 30-day and 12-month freedom from major adverse events (amputation, death, TLR/TVR (target vessel revascularisation)), MI (myocardial infarction), distal embolisation requiring intervention or hospitalisation, procedural success (≤30% residual diameter stenosis or occlusion after the procedure), improvement in Rutherford category (reduction ≤1 category), and improvement in ABI (increase ≥0.10 from baseline). Preliminary results from the first six enrolled patients indicate the safety and feasibility of sirolimus-coated balloon angioplasty [38].

The SIRONA trial is a single-blinded, multicentre, randomised controlled noninferiority trial comparing sirolimus-coated to commercially available paclitaxel-coated balloon angioplasty with over 2 years of follow-up. It enrolled 478 participants with symptomatic femoropopliteal artery disease (Rutherford category 2 to 4) due to stenosis or restenosis. After pre-dilation, participants are randomly assigned in a 1:1 ratio to receive either sirolimus- or paclitaxel-coated balloon angioplasty. Primary noninferiority endpoints include primary patency and a composite of all-cause mortality, major target limb amputation, and clinically driven target lesion revascularisation at 12 months. Secondary outcomes include clinical improvement, changes in quality of life, and safety over 60 months [39].

On 29 May 2023, Concept Medical Inc. (Tampa, FL, USA) received FDA approval (IDE) to study its Magic Touch PTA sirolimus-coated balloon for treating peripheral artery disease in the superficial femoral artery (SFA). This approval enables the launch of a pivotal clinical trial to demonstrate the device’s safety and effectiveness in the femoral and popliteal segments [40].

### 1.6. SELUTION SLR

The SELUTION SLR (sustained limus release) system employs miniature reservoirs composed of a biodegradable polymer infused with sirolimus at a dosage of 1 μg/mm^2^. These micro-reservoirs are coated onto the balloon using a proprietary amphipathic transfer membrane, which safeguards them throughout the processes of balloon insertion, crossing the lesion, and inflation. Upon balloon inflation, the transfer membrane, housing the micro-reservoirs, disengages from the balloon’s surface and adheres to the vessel’s inner lining, ensuring a consistent and prolonged release of sirolimus into the vessel wall [41].

The SELUTION SLR FIH trial was the first to assess the SELUTION SLR drug-eluting balloon (DEB) for femoropopliteal lesions in fifty claudicant patients at four German centres, with up to 34% having significant calcifications. The primary endpoint at 6 months was SFA angiographic late lumen loss (LLL), which had a median of 0.19 mm (range −1.16 to 3.07) and a mean of 0.29 ± 0.84 mm. Primary patency by duplex ultrasound was 88.4%, and freedom from angiographic binary restenosis was 91.2%. Over a 6-month period, there were significant improvements in Rutherford categories (*p* < 0.001) and ABI measurements (*p* < 0.001), with no major adverse events [41].

The SUCCESS PTA study is a real-world, prospective, multicentre, single-arm post-market surveillance study of the SELUTION SLR 018 sirolimus-eluting balloon for treating de novo/restenotic lesions in the SFA, popliteal, tibial, and pedal arteries. It aims to recruit at least 722 subjects from 50 sites across Europe, Asia, Canada, and South America, with no specific patient selection criteria. The primary study goal is to assess freedom from CD-TLR at one year. Major secondary endpoints include evaluating device and procedural success, major adverse limb events (MALE) such as severe limb ischemia leading to intervention or major vascular amputation (above the ankle), as well as assessing mortality, amputation, changes in the Rutherford score, alterations in ankle–brachial index, and wound status where applicable. Follow-up assessments are planned at one, six, and 12 months initially, and then annually for up to five years [42].

The Limus Flow study is an investigation focusing on vascular and endothelial dysfunction, a key factor in atherosclerosis development and cardiac events. Procedures such as percutaneous transluminal angioplasty (PTA) and drug-coated balloon (DCB) treatment affect endothelial balance, but the precise mechanisms are not fully understood. This study aims to assess the impact of novel devices, such as SELUTION SLR, on vascular function in infrainguinal arteries. The trial plans to enrol 70 participants, randomly assigning them in a 1:1 ratio to receive SELUTION SLR^TM^ (*n* = 35) or an active comparator, Medtronic InPact, a Paclitaxel-eluting balloon (*n* = 35). The primary outcome measures change in flow-mediated dilation (FMD) of the nonstenotic proximal SFA segment at 1, 6, and 12 months after the procedure [43]. SELUTION4SFA is a randomised trial set to assess the safety and effectiveness of SELUTION SLR^TM^ 018DEB (Med. Alliance, SA, Mont-sur-Rolle, Switzerland) in comparison to plain balloon angioplasty for treating PAD in the SFA and proximal popliteal artery (PPA). The study aims to recruit 300 patients from over 30 U.S. centres and 10 international centres. The primary effectiveness measure is the 12-month primary patency of the target lesion, while the primary safety measure is the absence of death within 30 days [44].

The PRISTINE TRIAL is a clinical study assessing the safety and effectiveness of treating TASC C and D atheroma-occlusive infra-inguinal disease in chronic limb-threatening ischemia patients from Singapore using the Selution Sirolimus Drug-Coated Balloon. The 12-month follow-up results, presented at CIRSE 2023 in Copenhagen, Denmark, reveal a primary patency rate of 69.7% at 6 months and 57.1% at 12 months. TLR rates were 15.2% at 6 months and 21.4% at 12 months, while amputation-free survival rates stood at 83.6% at 6 months and 72.3% at 12 months. An overall improvement in the Rutherford category compared to baseline was observed in 83% of patients [45,46].

### 1.7. Other Trials

#### 1.7.1. CVT-SFA Trial

The CVT-SFA First in Human Trial aimed to assess the prevention of restenosis using the CVT everolimus-coated PTA Catheter for treating femoropopliteal artery lesions. It was a prospective, multi-centre, open, single-arm study involving 75 participants with specific artery lesions (excluding in-stent cases) up to 150 mm in length. The study compared the CVT everolimus-coated PTA catheter to predefined Objective Performance Goals (OPG) for safety and effectiveness based on historical data (70% and 65%, respectively). Primary outcomes included the rate of MAE, a composite of cardiovascular death, limb amputation, and ischemia-driven TLR at 6 months post-procedure, and patency at 6 months post-procedure, defined as freedom from restenosis by duplex ultrasonography (PSVR ≤ 2.4 or ≤50% stenosis) and freedom from ischemia-driven TLR. The study achieved 97.3% for MAE and 92.6% for patency at 6 months, surpassing the predefined OPG of 70% and 65%, respectively [47].

#### 1.7.2. PREVISION Trial

The PREVISION FIH trial is a non-randomised, single-arm, prospective, multicentre study designed to assess the safety of BD Interventional SCBS for treating peripheral artery disease (PAD) in the femoropopliteal arteries. The study plans to include 50 participants among centres in Australia, New Zealand, and Singapore. The primary outcome measure focuses on determining LLL at 6 months using quantitative vascular angiography [48].

## 2. Discussion

### 2.1. Drug-Eluting Stent

The IMPERIAL trial, which compared two paclitaxel-eluting stents (Eluvia [Boston Scientific] with a polymer coating and Zilver PTX [Cook Medical, Queensland, Australia] without), the primary patency rate after 24 months of follow-up was 83.0% for Eluvia and 77.1% for Zilver PTX (Müller-Hülsbeck et al. 2021) [49].

The FIH MAJESTIC trial evaluated the outcomes of Eluvia stents [Boston Scientific, Mascot, Australia] in 57 patients with leg artery issues (Rutherford 2, 3, or 4) and found that this paclitaxel-eluting stent showed good long-term results: 83.5% primary patency at 24 months, 92.8% and 85.3% freedom from target lesion revascularisation (TLR) at 24 months and 36 months, respectively. No major amputations (MA) nor stent fractures were reported (Müller-Hülsbeck et al. 2017) [50].

In the BATTLE trial, a head-to-head randomised comparison of Zilver PTX and a bare metal stent (BMS), patency rates at 2 years were 78.8% for the drug-eluting stent (DES) and 74.6% for the BMS (Gouëffic et al. 2020) [51].

The EMINENT trial, a prospective, randomised 2:1, controlled, multicentre European study, comparing the Eluvia (Boston Scientific) DES and a BMS, reported patency rates at 12 months were, respectively, 83% and 76.3% (Gouëffic et al. 2022) [52].

Additionally, the ILLUMINA study reported a low 6.9% rate of CD-TLR after 2 years, which is the lowest rate observed so far in femoropopliteal interventions (Steiner et al. 2022) [53].

In the IMPERIAL trial, the 2-year TLR rates were 12.7% for Eluvia and 20.1% for Zilver PTX. In the BATTLE trial, the 2-year TLR rates were 12.4% for the Eluvia DES and 14.4% for the BMS. In the EMINENT trial, the TLR was the same for both groups (11.8%). In the Majestic trial, TLR was 7.2% and 14.7% at 24 months and 36 months, respectively (Gouëffic et al. 2020, 2022; Müller-Hülsbeck et al. 2017, 2021) [49,50,51,52].

Considering the differences between the trials, the results of the ILLUMINA study appear very promising (Table 1).

The ILLUMINA trial demonstrated superior efficacy compared to the IMPERIAL trial, with a notably lower 2-year CD-TLR rate (6.9% vs. 12.7% for Eluvia). This suggests that the sirolimus-based NiTiDES stent in ILLUMINA may offer advantages over paclitaxel-based platforms used in IMPERIAL (Müller-Hülsbeck et al. 2021) [49]. Notably, the inclusion of a significant diabetic cohort (35%) in ILLUMINA revealed favourable outcomes for amphilimus-eluting technology, supporting its potential benefit in diabetic patients historically less responsive to paclitaxel or mTOR inhibitor stents. These findings underscore the evolving role of targeted drug-eluting stent selection—particularly amphilimus-based devices—for optimising femoropopliteal interventions, especially in high-risk subgroups. Further research should directly compare the NiTiDES stent to other commonly used DES in clinical practice to determine its role in treating SFA lesions (Steiner et al. 2022; Byrne et al. 2017; Romaguera et al. 2016) [53,54,55].

The evolution of drug-eluting stent technology for the treatment of peripheral artery disease has shown progressive improvement in outcomes. The polymer-based Eluvia stent consistently demonstrates superior performance compared to both bare metal stents and the non-polymer Zilver PTX. The promising results from the ILLUMINA study suggest that newer drug formulations such as amphilimus may further advance treatment options, particularly for challenging patient populations such as diabetics. Further head-to-head comparisons between the NiTiDES stent and other commonly used DES would help clarify its optimal role in treating superficial femoral artery lesions and determine whether its promising early results translate to sustained long-term benefits.

### 2.2. Bioresorbable Vascular Scaffold

In the context of resorbable stents, the results reported in this BVS I study are the highest among the available studies in the literature (Table 2). Still, the company decided not to continue research in this field (Lammer et al. 2016) [32]. All the studies listed in the table regarding the bioresorbable scaffolds with or without drug release have been discontinued (Bontinck et al. 2016; Bradbury et al. 2010; Werner et al. 2014) [56,57,58].

The Efemoral device (Efemoral Medical, Inc, Los Altos, CA, USA) is an innovative, resorbable scaffold designed for femoropopliteal percutaneous intervention. It comprises 10 mm polylactide-based stents arranged sequentially on a 60 mm angioplasty balloon, with spaces between them to allow artery flexibility. In a preclinical study, using these short, serial scaffolds in long-segment porcine peripheral artery stenting did not affect the vessel’s diameter or cause crushing, crimping, or compression. The scaffolds maintained consistent wall apposition without fracture or deformation, but the spaces between them shortened significantly (Byrne et al. 2017; El Khoury et al. 2022) [55,59].

The experimental device is in its first-in-human clinical trial (FIH) (ClinicalTrials.gov identifier, NCT04584632), named Efemoral I. This prospective, single-arm, open-label, multicentre study plans to enroll 100 participants. The device will be implanted in vessels with diameters between 5.5 mm and 6.5 mm and lesion lengths of 90 mm or less. The primary study endpoints are Major Adverse Events (MAE) at 30 days and freedom from binary restenosis at 12 months [60].

The discontinuation of bioresorbable scaffold trials stems from a combination of safety concerns, efficacy limitations, and commercial factors rather than any single issue. Despite the theoretical advantages of temporary scaffolding that eventually disappears, first-generation BRS technology faced significant hurdles in matching the performance of modern metallic DES. The femoral device represents a novel approach to addressing these challenges, specifically for peripheral applications. The outcomes of the Femoral I trial will be crucial in determining whether this new design concept can overcome the limitations that led to the discontinuation of previous bioresorbable scaffold programs.

### 2.3. Drug-Eluting Balloon

The 6-month primary patency (88.2%) for femoropopliteal lesions with the MagicTouch PTA SCB in this study is comparable to the SELUTION SLR SCB femoropopliteal study (88.4%) [31]. It also aligns with paclitaxel-coated balloons (PCB) in previous studies, with 6-month primary patency rates of 87% and 90% reported in RANGER SFA and LEVANT 2, respectively. For PCBs, 12-month primary patency was 86.4% in RANGER SFA (Sachar et al. 2021) [61] and 65.2% in LEVANT 2 (Rosenfield et al. 2015) [62] while Magic Touch SCB reported a 78.6% primary patency [24]. The newly released data at 24 months demonstrated a drop in primary patency for SCB (53%) (Chek Choke, Yap, and Chong 2023) [34]. The same trend was observed in COMPARE TRIAL, reporting a primary patency of 66.7% and 65.5% at 24 months for IN.PACT and RANGER, respectively (Steiner et al. 2022) [53]. In the PACT SFA trial, the primary patency rate at 24 months was 78.9% (Micari et al., 2018) [63]. As regards TLR, those reported by the XTOSI trial remain very promising and sustained over the 24-month follow-up (Table 3). When examining the device’s performance, the results are in line with those published in previous studies.

The findings from SELUTION SLR FIH remained consistent throughout the 24-month follow-up period, as demonstrated by T. Zeller at VIVA 2019, Las Vegas, USA, where a primary patency rate of 81.6% and a TLR rate of 4.3% were reported. There were no reported deaths or instances of minor or major amputations. The authors also noted a continuous and lasting improvement in ABI and walking capacity. Furthermore, during the Japan Endovascular Treatment (JET) conference in Tokyo, Japan, in May 2023, the preliminary 12-month follow-up results of the “SELUTION SFA Trial—Japan” were presented. This prospective, multicentre, single-arm study involved 134 patients across 13 Japanese sites with de novo and nonstented restenotic lesions in the SFA/popliteal artery. The primary endpoint, 12-month primary patency, was reported at 87.9%, with a 97% freedom from TLR rate, a 0.7% thrombosis rate, and no major amputations or deaths. The authors noted that these outcomes were consistent with those previously reported by T. Zeller at VIVA 2019 (Anon n.d.-b; Iida et al. 2024) [64,65].

Among the published studies (Table 3), the result of the SELUTION SLR FIH trial seems to be very encouraging, reporting the highest primary patency and the lowest TLR up to 24 months follow-up [64].

However, comparisons with other trials should be made cautiously and serve as a general reference.

The comparative analysis of sirolimus and paclitaxel DCBs reveals a complex interplay of factors affecting performance. While most technologies show declining efficacy over time, the SELUTION SLR demonstrates unusually durable results at 24 months. This suggests that drug delivery technology may be as important as the choice of drug itself. The significant drop in primary patency for MagicTouch SCB at 24 months highlights the challenge of maintaining long-term efficacy with balloon-delivered drugs. Future developments should focus on extending drug retention time and addressing the biological mechanisms of late failure. When interpreting these results, it is essential to consider that direct cross-trial comparisons have limitations due to differences in study design, patient populations, and endpoint definitions. Head-to-head randomized trials would provide more definitive evidence of comparative efficacy between these technologies.

## 3. Future Directions

Modern devices provide increasingly refined control mechanisms, and the ideal platform should deliver pharmacological agents consistently throughout its functional lifespan, ensure a predictable therapeutic endpoint, promote effective re-endothelialisation, and achieve uniform drug distribution within the vessel. Achieving optimal interventional results is contingent on a nuanced understanding of the interdependence between device mechanisms, delivery mode, and vascular biology; lesion morphology, in particular, critically influences drug retention and local response, thereby shaping current and future paradigm shifts in interventional strategies.

Within cardiology, the broader integration of advanced intravascular imaging modalities, such as intravascular ultrasound (IVUS) and optical coherence tomography (OCT), has greatly facilitated lesion visualisation and post-procedural assessment; when these tools are combined with virtual histology (VH), there is further enhancement in lesion classification, enabling clinicians to tailor therapeutic strategies more precisely to individual lesion characteristics and thus optimize outcomes [66].

Accordingly, clinical recommendations must remain provisional: although sirolimus-eluting devices currently show considerable potential based on mid-term data, their widespread routine use should be withheld until their benefits are confirmed by further high-quality randomised controlled trials (RCTS) and long-term evidence, ideally reinforced by the utilisation of advanced imaging modalities. Active engagement in rigorous clinical trials remains crucial for closing existing knowledge gaps and informing the development of evidence-based, patient-centred vascular care.

## 4. Conclusions

The introduction of sirolimus-based drug-eluting stents and balloons has markedly advanced the ongoing evolution of endovascular therapy for femoropopliteal artery disease. Emerging clinical data suggest that these devices offer significant benefits over established paclitaxel-based technologies, particularly in terms of improved drug distribution, an enhanced therapeutic window, and a favourable safety profile. Despite these encouraging findings, the body of evidence remains limited, and several challenges and questions persist regarding long-term efficacy, safety, and the comparative performance of these innovative devices. Given these uncertainties, it is essential to conduct rigorous long-term follow-up studies to understand better the durability of clinical outcomes, potential late adverse effects, and the real-world effectiveness of limus-based technologies across diverse patient populations. Such longitudinal data will be critical in informing clinical best practices and in guiding the selection of the most appropriate therapies for individuals with symptomatic femoropopliteal disease.

Furthermore, direct head-to-head randomised controlled trials are required to establish clear comparative benchmarks between available drug-eluting platforms and to delineate their relative efficacy, safety, and cost-effectiveness. These studies will not only help identify the ideal device for various clinical scenarios; they will also clarify the precise role that each technology should play in managing symptomatic patients. In summary, sirolimus-based drug-eluting devices represent a significant step forward in the management of femoropopliteal artery disease. However, their incorporation into standard clinical algorithms should be predicated on robust evidence from extended follow-up and high-quality comparative trials. Continued research is paramount to realise the potential of these technologies fully and to optimise outcomes for patients with peripheral arterial disease.

## Figures and Tables

**Figure 1 jcm-14-03594-f001:**
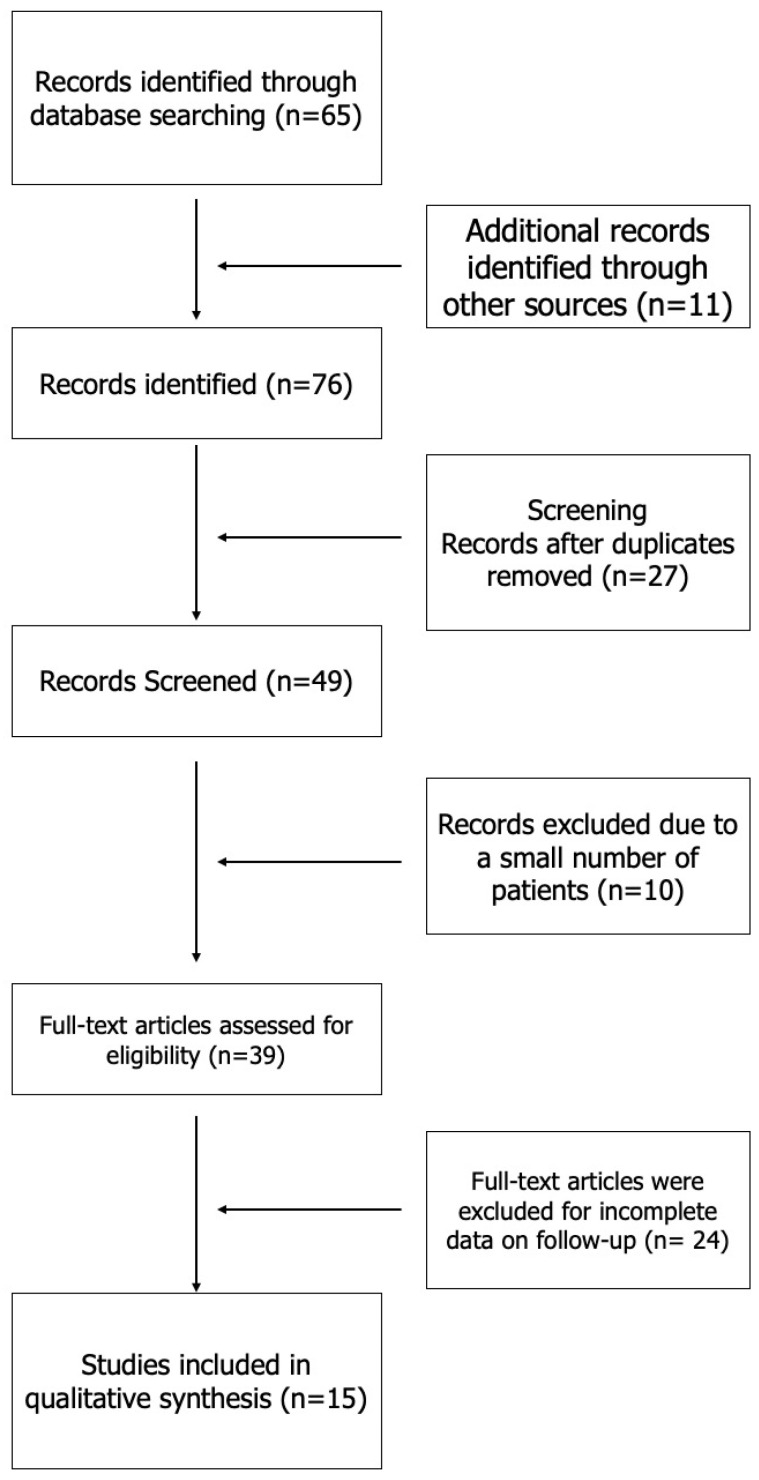
PRISMA flowchart of study selection for the review of epidemiology and interventions in femoropopliteal peripheral artery disease.

**Table 1 jcm-14-03594-t001:** Principal trials with 24 months follow-up evaluating the primary patency and target lesion revascularisation.

	No. of Patients	Primary Patency (24 mos)	TLR (24 mos)
NiTiDES (ILLUMINA Trial) [53]	100	83.4%	6.9%
Eluvia (IMPERIAL Trial) [49]	309/465	83.5%	7.2%
Zilver PTX (IMPERIAL Trial) [49]	156/465	77.1%	20.1%
Eluvia (MAJESTIC Trial) [50]	57	83.5%	7.2%
Zilver PTX (BATTLE Trial) [51]	90/181	78.8%	12.4%
BMS (BATTLE Trial) [51]	91	74.6%	14.4%
Eluvia (EMINENT Trial) [52]	508/775	83% (12 mos)	11.85% (12 mos)
BMS (EMINENT Trial) [52]	267/775	76.3% (12 mos)	11.8% (12 mos)

No. = number; mos = months; TLR = target lesion revascularisation.

**Table 2 jcm-14-03594-t002:** Principal trials evaluating the outcomes of bioresorbable scaffolds.

	No. of Patients	Primary Patency		TLR	
		12 (mos)	24 (mos)	12 (mos)	24 (mos)
ESPRIT—I BVS [32]	35	87.9%	83.9%	8.8%	11.8%
REDEMY [56]	99	58%	/	33%	/
BASIL vs. CFE—BASI [57]	40/80	80%	/	12.4%	/
BASIL vs. CFE—Sugery [57]	40/80	100%	/	14.4%	/
GAIA [58] study	30	32.1% (12 mos)	/	57.1% (12 mos)	/

TLR = target lesion revascularization; mos = months.

**Table 3 jcm-14-03594-t003:** Principal trials with a 24-month follow-up evaluating primary patency and target lesion revascularisation.

			Primary Patency			TLR		
	No. of Patient	Lesion Length	6 mos	12 mos	24 mos	6 mos	12 mos	24 mos
XTOSI [33,44]	20/50	227 ± 108 mm	88.2%	78.6%	53%	5.6%	5.9%	11% (overall)
SELUTION SFA FIM [64]	50	64.3 ± 42.8 mm	88.4%	75.7%	81.6%	2.3%	4.3%	4.3%
COMPARE/IN.PACT [53]	207/414	128.3 ± 97.3 mm	/	81.5%	66.7%	2.1%	7.4%	13%
COMPARE/RANGER [53]	207/414	123.9 ± 97.8 mm	/	83%	65.5%	2.5%	9.5%	17.3%
RANGER SFA [61]	71/105	68 ± 46 mm	87%	86.4%	/	5.6%	8.8%	/
LEVANT 2 [62]	316/476	107.9 ± 47.8 mm	90%	65.2%	/	/	12.1%	/
IN.PACT SFA [63]	220/331	89.4 ± 48.9 mm	/	82.2%	78.9%	/	2.4%	9.1%

No. = number; mos = months; TLR = target lesion revascularisation.

## Data Availability

No new data were created or analysed in this study.

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
