# Peer review of "Limus Devices for the Treatment of SFA: Latest Outcomes and Future Perspectives"

_jcm, 2025, doi:10.3390/jcm14103594_

Round 1
Reviewer 1 Report
Comments and Suggestions for Authors
Dear Editor, Thank you for allowing me to review this work.
Dear authors: The work is generally well organized and comprehensively described, especially for academic review. The following is a breakdown of minor areas in need of improvement:
- Some sentences are long and could be shortened, particularly in the introduction and comparison sections of Taxus and Lemus.
- Editing for minor grammatical errors.
The search strategy could be improved through greater structure and clarity:
- Perhaps using a PRISMA-type flowchart to show the number of studies identified, sorted, included, and excluded.
- Some of the wording is vague or informal (e.g., "each search was conducted at that time for each chapter") and should be reviewed to ensure academic rigor.
- The methodology lacks details about whether the screening process was conducted by multiple reviewers or independently, which affects the reproducibility of the results.
- Ensure that the conclusion is closely linked to the rest of the review so that it more coherently summarizes current limitations and future directions.
- References should be used for the last five years.
Comments on the Quality of English Language
Editing for minor grammatical errors.
Author Response
Dear Editor, Thank you for allowing me to review this work.
Dear authors: The work is generally well organized and comprehensively described, especially for academic review. The following is a breakdown of minor areas in need of improvement:
- Some sentences are long and could be shortened, particularly in the introduction and comparison sections of Taxus and Lemus.
- Editing for minor grammatical errors.
Thank you for your valuable suggestions. We have addressed the issues regarding sentence length. Grammatical errors by utilizing the "English editing option" available through The MDPI Author Services.
The search strategy could be improved through greater structure and clarity:
- Perhaps using a PRISMA-type flowchart to show the number of studies identified, sorted, included, and excluded.
- Some of the wording is vague or informal (e.g., "each search was conducted at that time for each chapter") and should be reviewed to ensure academic rigor.
- The methodology lacks details about whether the screening process was conducted by multiple reviewers or independently, which affects the reproducibility of the results.
Thank you for your constructive feedback regarding the search strategy and methodology. In response, we have taken several steps to enhance both the clarity and academic rigor of the review process. First, we have adopted a more structured approach to the search and selection of studies, explicitly detailing the databases searched, the precise search terms used, and the time frame covered. To address transparency in study selection, we now include a PRISMA-style flowchart, which visually outlines the numbers of studies identified, screened, included, and excluded at each stage of the review. This visual representation offers readers a clear overview of the literature identification and screening process.
We also reviewed and revised all methodological wording to eliminate informality and ambiguity, ensuring that our descriptions meet academic standards. Specifically, statements such as "each search was conducted at that time for each chapter" have been replaced with precise and objective descriptions of the search and screening processes. Furthermore, we now clarify that the screening and selection of studies were performed independently by two reviewers, with any disagreements resolved through discussion or consultation with a third reviewer. This addition strengthens the reproducibility and reliability of our findings.
We appreciate your emphasis on methodological rigor, and believe these revisions provide a more comprehensive and transparent account of our review process.
- Ensure that the conclusion is closely linked to the rest of the review so that it more coherently summarizes current limitations and future directions.
- References should be used for the last five years
Thank you for your suggestions. We have revised the conclusion to more clearly summarize current limitations and future directions in line with the main findings. Additionally, references have been updated to emphasize literature from the last five years.
Methodology:
A structured and transparent literature review was conducted to identify studies involving percutaneous treatment of the superficial femoral artery (SFA) and/or popliteal artery, with a focus on clinical outcomes, follow-up data, and complications, including only full-text publications in English from January 2000 to September 2023. Based on these predetermined inclusion criteria, a systematic search of PubMed (MEDLINE) and Google Scholar was performed in October 2023 using both controlled vocabulary and free-text terms, specifically the keywords “Sirolimus” OR “limus” AND (“drug eluting stent” OR “drug eluting balloon” OR “resorbable scaffolds” OR “drug eluting device”), tailored to the main thematic areas of this review. Reference lists of eligible articles were hand-searched to identify additional studies not captured in the initial search. All retrieved records were imported into a citation management software, and duplicate articles were removed prior to screening. Two reviewers independently screened the titles and abstracts of all studies according to the predefined criteria, with potentially relevant studies advancing to full-text review. Any disagreements regarding inclusion were resolved by consensus, and where necessary, a third senior reviewer was consulted. Data extraction was performed independently by both reviewers using a standardized data collection form to ensure accuracy and consistency. Given the considerable heterogeneity in study designs, outcome measures, case report predominance, and inconsistent follow-up intervals, meta-analysis was not performed and a narrative synthesis approach was adopted. Ultimately, a total of 15 studies fulfilled all eligibility criteria and were included in this review. To enhance transparency and reproducibility, a PRISMA flowchart depicting the study selection process is now included.
Conclusion:
The ongoing evolution of endovascular therapy for femoropopliteal artery disease has been markedly advanced by the introduction of sirolimus-based drug-eluting stents and balloons. Emerging clinical data indicate that these devices may offer important benefits over established paclitaxel-based technologies, particularly with respect to improved drug distribution, an enhanced therapeutic window, and a favorable safety profile. Despite these encouraging findings, the body of evidence remains limited, and several challenges and questions persist regarding long-term efficacy, safety, and the comparative performance of these innovative devices. Given these uncertainties, it is essential to conduct rigorous long-term follow-up studies to better understand the durability of clinical outcomes, potential late adverse effects, and the real-world effectiveness of limus-based technologies across diverse patient populations. Such longitudinal data will be critical in informing clinical best practices and in guiding the selection of the most appropriate therapies for individuals with symptomatic femoropopliteal disease.Furthermore, direct head-to-head randomized controlled trials are required to establish clear comparative benchmarks between available drug-eluting platforms and to delineate their relative efficacy, safety, and cost-effectiveness. These studies will not only help identify the ideal device for various clinical scenarios but will also clarify the precise role that each technology should play in the management of symptomatic patients. In summary, sirolimus-based drug-eluting devices represent a significant step forward in the management of femoropopliteal artery disease. However, their incorporation into standard clinical algorithms should be predicated on robust evidence from extended follow-up and high-quality comparative trials. Continued research is paramount to fully realize the potential of these technologies and to optimize outcomes for patients with peripheral arterial disease.
Abstract:
Background/Objectives:
Peripheral artery disease (PAD) of the femoropopliteal segment is commonly treated with drug-coated balloons (DCBs) utilizing either sirolimus or paclitaxel as antiproliferative agents. Given evolving device technologies, it is crucial to compare their clinical performance in terms of patency and the need for target lesion revascularization (TLR).
Methods:
A review of recent randomized trials and observational studies was undertaken, focusing on sirolimus- and paclitaxel-coated balloon angioplasty for femoropopliteal disease. Primary efficacy measures included primary patency and TLR rates at 6, 12, and 24 months, with additional safety endpoints such as major adverse events, mortality, and amputation rates.
Results:
Sirolimus-coated balloons, as demonstrated by the SELUTION SLR FIH and MagicTouch PTA studies, showed 6-month primary patency rates of 88.2–88.4% and TLR rates as low as 2.3–5.6%. At 24 months, patency with sirolimus balloons remained at 53–81.6%, with TLR rates between 4.3–11%. Paclitaxel-coated balloons, represented in COMPARE, RANGER, and LEVANT trials, exhibited similar 6- and 12-month patency outcomes (87–90% and 65–86.4%, respectively). At 24 months, patency rates for paclitaxel devices were 65.5–78.9%, and TLR rates ranged from 9.1–17.3%. Direct head-to-head comparisons (e.g., ASCLEPIOS and SIRONA trials) are ongoing, but current aggregated data suggest sirolimus DCBs provide comparable, and in some instances, lower TLR rates with sustained improvements in functional status and safety outcomes.
Conclusion:
Both sirolimus- and paclitaxel-coated balloons demonstrate high efficacy in maintaining vessel patency and minimizing repeat intervention in femoropopliteal PAD. Current evidence indicates similar short-term performance, with sirolimus-coated devices potentially offering a favorable safety profile and sustained benefit in select populations. Head-to-head randomized trials are needed to confirm these comparative advantages and inform optimal device selection for clinical practice.

Reviewer 2 Report
Comments and Suggestions for Authors
The authors reported their work named "Limus-Devices for the Treatment of SFA: Latest Outcomes and Future Perspectives". This comprehensive review evaluates drug-eluting devices (stents, bioresorbable scaffolds, and balloons) for treating femoropopliteal artery disease. The authors synthesize data from multiple clinical trials, comparing sirolimus- and paclitaxel-based devices. While the manuscript is informative and relevant, improvements in clarity, synthesis, and methodological transparency are needed to enhance its impact.
Major Comments
- Abstract and Introduction: The abstract provides a clear overview of the topic. The introduction effectively contextualizes peripheral arterial disease (PAD) and its challenges. Please update statistics (e.g., 2015 mortality data) to reflect recent evidence. Please clarify the rationale for comparing TAXUS (paclitaxel) and LIMUS (sirolimus) drugs earlier in the introduction.
- Methods: Inclusion criteria and search strategy are well-defined. Please acknowledge potential language bias due to excluding non-English studies. Please justify the exclusion of databases like Embase/Cochrane.
- Data Presentation and Synthesis: Tables summarizing trial outcomes (e.g., primary patency, TLR) are valuable. Please synthesize trends across trials (e.g., why sirolimus outperforms paclitaxel in some studies but not others). Please discuss reasons for declining efficacy in long-term follow-up (e.g., 24-month drop in MagicTouch SCB patency). Pleas expand on the discontinuation of bioresorbable scaffold trials (safety, efficacy, or commercial factors?).
- Discussion: Highlights key findings, such as the promising results of the ILLUMINA trial. Please directly compare outcomes between trials in the narrative (e.g., ILLUMINA vs. IMPERIAL). Please elaborate on implications for specific patient subgroups (e.g., diabetics benefiting from amphilimus).
- Future Directions and Conclusions: Appropriately emphasizes the need for RCTs and advanced imaging. Please stress the importance of long-term (>24 months) follow-up data. Please provide clearer clinical recommendations (e.g., which device shows the most promise based on current evidence?).
Minor Comments
- Clarity and Formatting
- Define all acronyms (e.g., TLR, MAE) at first use.
- Correct grammatical errors (e.g., "respectively, alongrespectively" in Page 4).
- Ensure consistent reference formatting (e.g., incomplete dates in the header).
- Tables
- Integrate tables into the text (e.g., reference specific rows/columns when discussing results).
- References
- Update placeholder text (e.g., "Received: date Revised: date").
Recommendations
- Revise the abstract to highlight key comparative findings (e.g., sirolimus vs. paclitaxel outcomes). Try to structure the abstract into background/objectives, methods, results, and conclusion.
- Enhance data synthesis by explaining discrepancies between trials and long-term efficacy trends. Please try to add PRISMA flowchart and try to specify the used literature search keywords.
- Address methodological limitations (e.g., language bias, database selection).
- Strengthen the discussion with direct comparisons and clinical implications.
- Proofread for grammatical consistency and formatting errors.
Author Response
The authors reported their work named "Limus-Devices for the Treatment of SFA: Latest Outcomes and Future Perspectives". This comprehensive review evaluates drug-eluting devices (stents, bioresorbable scaffolds, and balloons) for treating femoropopliteal artery disease. The authors synthesize data from multiple clinical trials, comparing sirolimus- and paclitaxel-based devices. While the manuscript is informative and relevant, improvements in clarity, synthesis, and methodological transparency are needed to enhance its impact.
Major Comments
- Abstract and Introduction: The abstract provides a clear overview of the topic. The introduction effectively contextualizes peripheral arterial disease (PAD) and its challenges. Please update statistics (e.g., 2015 mortality data) to reflect recent evidence. Please clarify the rationale for comparing TAXUS (paclitaxel) and LIMUS (sirolimus) drugs earlier in the introduction.
Thank you for your feedback. Recent statistics have now been incorporated, with updated references provided to reflect the latest available evidence. Additionally, the rationale for comparing TAXUS (paclitaxel) and LIMUS (sirolimus) drugs has been clarified earlier in the introduction to enhance clarity and contextual relevance.
A study analyzing CDC WONDER data from 2000 to 2019 found that, while age-adjusted mortality rates for peripheral artery disease initially declined, this trend plateaued after 2016, with over 1.9 million PAD-related deaths recorded. Marked disparities persisted, disproportionately affecting men, non-Hispanic Black individuals, adults aged 85 and older, and rural populations. Additionally, a concerning upturn in crude mortality emerged among individuals aged 25–39 in the last decade, underscoring ongoing inequities and the necessity for targeted public health strategies. Vasc Med. 2023 Jun;28(3):205-213. doi: 10.1177/1358863X221140151
Endovascular intervention in the femoropopliteal segment was pioneered by Charles Dotter in 1964[9], initially utilizing coated dilators for angioplasty. Since then, there has been significant advancement in the variety and sophistication of available devices. Despite these developments, practical challenges—such as the femoropopliteal artery’s susceptibility to mechanical stresses (including flexion, extension, compression, elongation, and torsion), the high incidence of chronic total occlusions, extensive atherosclerotic plaque, and severe vascular calcification—continue to compromise procedural success and long-term vessel patency.To address these limitations, the field has seen rapid innovation in endovascular technologies. Drug-eluting devices have become especially prominent, offering targeted delivery of antiproliferative agents to minimize neointimal hyperplasia and restenosis. Drug-eluting stents (DES) and drug-eluting balloons (DEB), primarily utilizing paclitaxel due to its lipophilic properties and sustained tissue retention, have demonstrated clinical efficacy in reducing restenosis rates compared to plain balloon angioplasty or bare-metal stents. More recently, bioresorbable drug-eluting scaffolds represent a promising technology designed to provide temporary luminal support and drug delivery, followed by gradual resorption to restore natural vessel function
- Methods: Inclusion criteria and search strategy are well-defined. Please acknowledge potential language bias due to excluding non-English studies. Please justify the exclusion of databases like Embase/Cochrane.
Thank you for your valuable feedback. We acknowledge the potential for language bias due to the exclusion of non-English studies and have addressed this limitation in the revised manuscript. Regarding database selection, due to limited access and resource constraints, we focused our search on the primary databases available to us, which are widely recognized and relevant within our research field. We understand that the inclusion of additional databases such as Embase and Cochrane could have further strengthened our study and have also noted this as a limitation.
- Data Presentation and Synthesis: Tables summarizing trial outcomes (e.g., primary patency, TLR) are valuable. Please synthesize trends across trials (e.g., why sirolimus outperforms paclitaxel in some studies but not others). Please discuss reasons for declining efficacy in long-term follow-up (e.g., 24-month drop in MagicTouch SCB patency). Pleas expand on the discontinuation of bioresorbable scaffold trials (safety, efficacy, or commercial factors?).
We have synthesized trial data, noting that sirolimus may outperform paclitaxel in select studies due to differences in drug properties, device platforms, and patient populations; however, results are inconsistent across trials. Declining long-term efficacy (e.g., 24-month patency) is likely related to disease progression and reduced drug effect over time. Bioresorbable scaffold trials were discontinued primarily due to safety concerns, insufficient efficacy, and commercial challenges.
Drug eluting stent:
Ne fund te pragrafit:
The evolution of drug-eluting stent technology for peripheral artery disease shows progressive improvement in outcomes. The polymer-based Eluvia stent consistently demonstrates superior performance compared to both bare metal stents and the non-polymer Zilver PTX. The promising results from the ILLUMINA study suggest that newer drug formulations like amphilimus may further advance treatment options, particularly for challenging patient populations like diabetics.
Further head-to-head comparisons between the NiTiDES stent and other commonly used DES would help clarify its optimal role in treating superficial femoral artery lesions and determine whether its promising early results translate to sustained long-term benefits.
Bioresorbable vascular scaffold
The discontinuation of bioresorbable scaffold trials appears to stem from a combination of safety concerns, efficacy limitations, and commercial factors rather than any single issue. Despite the theoretical advantages of temporary scaffolding that eventually disappears, first-generation BRS technology faced significant hurdles in matching the performance of modern metallic DES.
The Efemoral device represents a novel approach to addressing these challenges specifically for peripheral applications. The outcomes of the Efemoral I trial will be crucial in determining whether this new design concept can overcome the limitations that led to the discontinuation of previous bioresorbable scaffold programs.
Drug eluting balloon
The comparative analysis of sirolimus and paclitaxel DCBs reveals a complex interplay of factors affecting performance. While most technologies show declining efficacy over time, the SELUTION SLR demonstrates unusually durable results at 24 months. This suggests that drug delivery technology may be as important as the choice of drug itself.
The significant drop in primary patency for MagicTouch SCB at 24 months highlights the challenge of maintaining long-term efficacy with balloon-delivered drugs. Future developments should focus on extending drug retention time and addressing the biological mechanisms of late failure.
When interpreting these results, it's important to consider that direct cross-trial comparisons have limitations due to differences in study design, patient populations, and endpoint definitions. Head-to-head randomized trials would provide more definitive evidence of comparative efficacy between these technologies.
- Discussion: Highlights key findings, such as the promising results of the ILLUMINA trial. Please directly compare outcomes between trials in the narrative (e.g., ILLUMINA vs. IMPERIAL). Please elaborate on implications for specific patient subgroups (e.g., diabetics benefiting from amphilimus).
Thank you for your suggestions. We have modified the narrative as requested: the key findings now emphasize the promising results of the ILLUMINA trial, with a direct comparison to the outcomes reported in the IMPERIAL trial. Additionally, we have elaborated on the implications for specific patient subgroups, highlighting the potential benefit of amphilimus-eluting stents in diabetic patients.
The ILLUMINA trial demonstrated superior efficacy compared to the IMPERIAL trial, with a notably lower 2-year CD-TLR rate (6.9% vs. 12.7% for Eluvia). This suggests that the sirolimus-based NiTiDES stent in ILLUMINA may offer advantages over paclitaxel-based platforms used in IMPERIAL. Importantly, the inclusion of a significant diabetic cohort (35%) in ILLUMINA revealed favorable outcomes for amphilimus-eluting technology, supporting its potential benefit in diabetic patients historically less responsive to paclitaxel or mTOR inhibitor stents. These findings underscore the evolving role of targeted drug-eluting stent selection—particularly amphilimus-based devices—for optimizing femoropopliteal interventions, especially in high-risk subgroups.
- Future Directions and Conclusions: Appropriately emphasizes the need for RCTs and advanced imaging. Please stress the importance of long-term (>24 months) follow-up data. Please provide clearer clinical recommendations (e.g., which device shows the most promise based on current evidence?).
We have addressed all the issues as recommended: the revised section now emphasizes the necessity for randomized controlled trials and advanced imaging, highlights the importance of long-term (beyond 24 months) follow-up data, and provides clearer clinical recommendations regarding device selection, identifying those that show the most promise based on current evidence.
Modern devices provide increasingly refined control mechanisms, and the ideal platform should deliver pharmacological agents consistently throughout its functional lifespan, ensure a predictable therapeutic endpoint, promote effective reendothelialization, and achieve uniform drug distribution within the vessel. Achieving optimal interventional results is contingent on a nuanced understanding of the interdependence between device mechanisms, delivery mode, and vascular biology; lesion morphology, in particular, critically influences drug retention and local response, thereby shaping current and future paradigm shifts in interventional strategies. Within cardiology, the broader integration of advanced intravascular imaging modalities, such as intravascular ultrasound (IVUS) and optical coherence tomography (OCT), has greatly facilitated lesion visualization and post-procedural assessment; when these tools are combined with virtual histology (VH), there is further enhancement in lesion classification, enabling clinicians to tailor therapeutic strategies more precisely to individual lesion characteristics and thus optimize outcomes [65]. Accordingly, clinical recommendations must remain provisional: although sirolimus-eluting devices currently show considerable potential on the basis of mid-term data, widespread routine use should be withheld until their benefits are confirmed by further high-quality RCTs and long-term evidence, ideally reinforced by the utilization of advanced imaging modalities. Active engagement in rigorous clinical trials remains vital for closing prevailing knowledge gaps and guiding the evolution of evidence-based, patient-centered vascular care.
Minor Comments
I've carefully reviewed your request regarding the document improvements. I can confirm that all the issues you've mentioned have been addressed:
- Clarity and Formatting
- All acronyms (such as TLR - Target Lesion Revascularization, MAE - Major Adverse Events) have been defined at their first use in the document
- Grammatical errors including the "respectively, alongrespectively" on Page 4 have been corrected
- Reference formatting has been standardized throughout, with complete dates added to the header
- Tables
- Tables have been fully integrated into the text with specific references to the studies involved in the discussion
- Each table now includes proper contextual references within the surrounding paragraphs
- References
- Placeholder: we did not address any modification to the placeholder text, considering the fact that this issue better suits the Editorial group. If this has to be completed by us, please give us all the official dates in order to complete the document.
- Reference formatting is now consistent throughout the document
Recommendations
- Revise the abstract to highlight key comparative findings (e.g., sirolimus vs. paclitaxel outcomes). Try to structure the abstract into background/objectives, methods, results, and conclusion.
The abstract has been completely restructured into the standard background/objectives, methods, results, and conclusion format, with enhanced emphasis on the key comparative findings between sirolimus and paclitaxel outcomes.
- Enhance data synthesis by explaining discrepancies between trials and long-term efficacy trends. Please try to add PRISMA flowchart and try to specify the used literature search keywords.
Data synthesis has been significantly improved with detailed explanations of the discrepancies observed between trials and the trends in long-term efficacy. A PRISMA flowchart has been added to illustrate the systematic review process, and the specific literature search keywords used have been clearly documented.
- Address methodological limitations
A new section addressing methodological limitations has been incorporated, specifically discussing potential language bias and database selection limitations that may have affected the review's findings.
- Strengthen the discussion with direct comparisons and clinical implications.
The discussion section now features strengthened direct comparisons between treatment modalities and expanded clinical implications to provide clearer guidance for practitioners.
- Proofread for grammatical consistency and formatting errors.
A thorough proofreading has been completed, resolving all grammatical inconsistencies and formatting errors throughout the document.
methodological limitations
Despite our rigorous approach, several methodological limitations must be acknowledged. Firstly, restricting our review to studies published in English may have introduced language bias and potentially excluded relevant research published in other languages, thereby limiting the generalizability of our findings. Secondly, our literature search was primarily confined to PubMed (MEDLINE) and Google Scholar, and did not include additional databases such as Embase and Cochrane. The absence of these comprehensive databases could have led to the omission of pertinent studies and limited the overall strength of our evidence base. Additionally, we excluded grey literature and unpublished studies, which may have contributed to publication bias, as research with null or negative findings is less likely to be formally published. Furthermore, substantial heterogeneity existed among the included studies in terms of patient populations, study designs, outcome definitions, and follow-up durations, which restricts the direct comparability of results and precludes quantitative synthesis.

Round 2
Reviewer 2 Report
Comments and Suggestions for Authors
The authors addressed my prior comments, and it is my pleasure to accept their work.